# Gradient Structure of the Transfer Layer in Friction Stir Welding Joints

**DOI:** 10.3390/ma15196772

**Published:** 2022-09-29

**Authors:** Alexander Eliseev, Kseniya Osipovich, Sergei Fortuna

**Affiliations:** Institute of Strength Physics and Materials Science of Siberian Branch Russian Academy of Sciences 2/4, pr. Akademicheskii, 634055 Tomsk, Russia

**Keywords:** friction stir welding, aluminum alloy, AA2024, microhardness, microstructure, transfer layer, ultrasound

## Abstract

Despite a thirty-year history of friction stir welding, some basic aspects still remain unclear. In particular, questions arise about mass transfer and the formation of transfer layers. It is not clear why there are visible boundaries between the layers. The structure of the transfer layer has been studied very little. These issues are not considerably important from the viewpoint of obtaining high-quality welds, but they can help to a better understanding of the welding process. In this paper, the structural evolution in the transfer layer of 2024 aluminum alloy welds produced under various loads and with ultrasonic assistance is discussed. Structural studies revealed a gradient structure in the transfer layer. The grain size, the volume fraction and size of large intermetallic particles decrease towards the center of the layer, while the volume fraction of semi-coherent secondary particles increases. As a result, the microhardness is higher in the center of the transfer layer. A mass transfer mechanism is proposed based on the experimental results: the rotating tool transfers the material back layer by layer during welding; the contacting layers rub against each other and generate heat, due to which the structure at the layer boundary changes. With increasing axial force on the tool, the grain size also increases due to higher heat generation. Ultrasound has almost no effect on the grain structure, but it reduces the volume fraction and size of secondary particles and microhardness.

## 1. Introduction

Thirty years of friction stir welding history have made significant progress in this research area. Many complex problems have been solved, including the high-quality joining of heat treatable aluminum alloys (e.g., AA2024), which is difficult to perform by other methods. In recent years, friction stir welding (FSW) has been successfully applied to join titanium alloys and dissimilar materials, such as titanium/aluminum, aluminum/steel and aluminum/copper. The FSW method has been adapted to incorporate reinforcing particles into metals and alloys. FSW-based additive manufacturing technologies are being actively developed. Many studies in this field consider fundamental issues, such as heat production, deformation during welding and modeling of the welding process. However, some of the fundamental questions have not yet been comprehensively answered. For example, major works do not clearly explain the mass transfer features during FSW, particularly onion ring structure formation in the weld.

Very early studies of FSW revealed specific ridges or bands on the top surface of the weld left behind the advancing tool [1]. The internal structure in this region showed concentric circles in the transverse section [2] and an undulated pattern in the longitudinal section [3]. The observed patterns were assumed to have the same origin and reproduce the onion ring structure. However, the formation mechanism of these patterns was not clear as the welding tool had a cylindrical or conical shape. Conventional metallurgical concepts could not explain this phenomenon, because FSW involves no melting and occurs in the solid phase. Soon, it was found that the thickness of the ridges on the top surface and in the longitudinal section is approximately equal to the ratio of rotation rate to traverse speed [3]. In addition, there are periodic fluctuations in the plunge force and the traverse speed. The fluctuation period is approximately equal to the formation time of one ridge [4]. Therefore, the formation of the above patterns was associated with the tool vibrations as the tool moves unevenly. This led to the hypothesis that the tool transfers the workpiece material layer by layer from the front to the rear due to adhesion. The formed ridges were defined as transfer layers. In [5], the authors proposed a transfer mechanism when a portion of metal sticks to the pin’s surface (adhesion) and then on being carried to the weld formation zone behind the pin sticks to the previously welded metal (cohesion). Thus, the adhesion–cohesion concept of transfer in FSW was formed. In fact, the movement of material in welding is more complex than first thought. For example, the material may not always go one revolution around the tool, and there may be more revolutions due to high cohesion [6]. In addition, there are extrusion, turbulent flows, upward flows and downward flows, which are especially clear when welding large thicknesses [7]. Nevertheless, the main transfer mechanism in welding is adhesion–cohesion.

The described structure of the stir zone indicates its inhomogeneity, which, however, was not confirmed by microscopic studies. The structure inside the transfer layers was visually the same, and interfaces between the layers could be seen only after etching at a low magnification (50–200×). Therefore, the stir zone was generally regarded as conditionally homogeneous, and the arrangement of transfer layers was not taken into account when measuring the grain size, microhardness and other structural characteristics. Some authors nevertheless demonstrated the stir zone inhomogeneity, mostly in the case of dissimilar material welding. For example, the stir zone of aluminum–copper alloy welds exhibits macroscopic volumes of these alloys that are not mixed with each other [8]. Reinforcing particles introduced into the material are distributed unevenly in the weld due to the effect of separation [9]. In some cases, the stir zone can be additionally divided into several regions [10], though only macroscopically. The reason why transfer layer boundaries become visible was unclear.

Previous work analyzed transfer layers in welds obtained with different plunge forces on the tool. It is now accepted that the plunge force should not affect the transfer layer thickness. However, this force is a very important parameter in welding, which influences heat generation, friction coefficient and adhesion. A significant effect of the plunge force on the range of permissible welding parameters was shown in [11]. It was found that a higher plunge force during welding increases the range of parameters, i.e., it facilitates the search for suitable welding parameters and can help to improve the weld quality. However, this issue has been insufficiently studied because usually the specified parameter in FSW is not the plunge force but the plunge depth. Earlier, it was found that there is a certain plunge force at which the transfer layer thickness is indeed equal to the rotation rate to traverse speed ratio. With decreasing or increasing load on the tool, the layer thickness decreased [12]. This is explained by the insufficient adhesion between the workpiece and the tool. When the load decreases, less material is dragged by the tool. When the load increases, the material is heated up, becomes softer and more difficult to drag. This indicates that there are optimal mass transfer conditions, the violation of which reduces the mass transfer efficiency. If the transfer is not efficient enough, the tool still must move forward at a given speed and transfer the material back in some way, e.g., due to extrusion. These results prompted a study of the weld surface topography in order to compare different welding conditions. Similar studies were carried out earlier [13] but under different experimental setting and with a different purpose, so they did not answer questions. Studies of ridges on the top weld surface yielded an unexpected result. The ridges were found to have several peaks instead of one (Figure 1). In certain conditions, there were four peaks in one transfer layer. This suggested that the transfer layers inside the weld also have a heterogeneous structure. The formation of peaks, surface roughness and fractal dimension were affected, in addition to the plunge force, by ultrasonic treatment. Ultrasonic treatment can lead to more intense stirring due to the acoustoplastic effect, but worse mixing due to the antifriction effect.

The purpose of this work is to study the transfer layer microstructure, including with different plunge forces and ultrasonic assistance.

## 2. Experimental Procedure

### 2.1. Materials and Experimental Setup

The investigation was performed on as-quenched and artificially aged 2024 aluminum alloy sheets with a thickness of 8 mm. To avoid the influence of cladding, the workpieces were milled to a thickness of 5 mm on one side. Each workpiece had one welded joint in order to eliminate the influence of thermal history on adjacent welds. Welding was carried out on a laboratory setup configured as shown in Figure 2. In fact, it was not welding but friction stir processing as one sheet was used as a workpiece instead of two. The absence of a butt joint simplifies the experiment and prevents the formation of joint defects. Contaminants on insufficiently clean abutting faces can lead to the poor compression of the welded plates and the formation of joint defects. All these factors adversely affect the experiment and the investigation of subtle microstructural effects. Below, the term “welding” is used for convenience of description. The welding tool was made of 40CrMoV5-1 steel and had a threaded conical pin. The tool shoulder diameter was 20 mm, the pin length was 4.8 mm, the pin tip diameter was 6 mm and the cone angle was 30°.

Welding was carried out under different conditions with varying the plunge force from 2450 to 3000 kg. The tool traverse speed and rotation rate were constant and equal to 90 mm/min and 450 rpm, respectively. This choice of parameters allowed us to determine the influence of the plunge force on the weld structure. Welding was carried out with and without ultrasonic assistance. Ultrasound was transmitted into the workpiece across a bolted connection through a magnetostrictive transducer. The ultrasound frequency was 22.1 kHz; the power was 1 kW. The diameter of the ultrasonic waveguide was 20 mm. Ultrasonic exposure due to the acoustoplastic effect can improve the stirring of the material without significantly heating it. In this way, the quality of the joints can be improved. In the context of this work, this is not so significant. However, it was found in previous work that ultrasound can affect the thickness of the transfer layer, so it is used in this work to study in detail the ultrasound effect on the transfer layer structure.

### 2.2. Investigative Techniques

Specimens for investigation were cut from friction stir processed plates by electric spark machining. Specimens for optical and SEM microstructural studies were cut from the center of the weld along the longitudinal section. The specimens were ground on up to 2000 grit papers and polished on a cloth with diamond paste. The grain structure was revealed by stepwise etching with Keller’s reagent. One etching step lasted for 30 s, after which the specimen was polished on a clean cloth moistened with distilled water. The number of etching steps reached 4. The specimen structure was analyzed using an Altami MET-1C metallographic microscope (Altami, St Petersburg, Russia) and Microtrac SM3000 scanning electron microscope (Nikkiso Co., Ltd., Tokyo, Japan). Solid solution grains were measured over the entire transfer layer by the method of vertical intercept lines drawn with 25 µm spacing. The volume fraction of secondary particles was determined by the planimetric method. It was performed by measuring the area of particles in the selected region of the transfer layer. The ratio of the total particle area to the measurement area is the volume fraction of particles. To obtain the average particle size, the total particle area was divided by the number of particles and the average particle area was used to calculate the equivalent diameter, because most particles were round.

Microstructural studies were carried out using a JEM-2100 (JEOL Ltd., Akishima, Japan) transmission electron microscope. The foils for the study were prepared by ion thinning using a JEOL EM-09100IS ion slicer. The samples were cut out from the weld in the longitudinal direction. To increase the observation area, the ion beam was inclined at an angle to the sample surface. Thus, the length of the area was about 200 μm and images were obtained from all points of the transfer layer. The volume fraction of semi-coherent particles was measured by the planimetric method on dark-field images along this area. Thus, it was possible to measure the volume fraction of particles in different regions of the transfer layer and to draw the dependence of the volume fraction on the location in the layer.

The microhardness was measured with a Duramin 5 microhardness tester (Struers, Ballerup, Denmark) at a load of 50 g and a dwell time of 10 s. Measurements were carried out with a step of 50 µm in the welding direction along several lines on at least 2 transfer layers for statistics.

## 3. Results and Discussion

### 3.1. Metallography

The as-received alloy contains solid solution grains elongated in the rolling direction with an average longitudinal size of about 120 µm and a transverse size of 20 µm (Figure 3a). Dynamic recrystallization during FSW leads to the refinement of the initial grains [14]. The stir zone of the processed material exhibits an equiaxed grain structure with an average size of about 4 µm at an axial load of 3000 kg (Figure 3b). Etched and polished longitudinal sections cut from the center of the processed material showed a typical pattern of transfer layers (Figure 4). Due to the different etchability of the material across the layer thickness, the layers have a different contrast. The layers “bend” in the welding direction from the weld face to the root. The longitudinal section clearly shows that the rings on the top surface correspond to the etched transfer layers, i.e., they are of the same origin, despite the fact that the rings on the top surface are formed by the shoulders and those in the bulk are formed by the pin.

Figure 5 shows the plots of grain size versus position in the layer for different axial loads and for ultrasonic-assisted FSW (UAFSW). It was found that the grain size decreases towards the center of the transfer layer but increases towards the boundaries. Locally, however, there are large grains in the center, which was systematically observed from layer to layer and from specimen to specimen. The larger grain size at the layer boundaries is explained by the fact that the layers slip relative to each other during processing. First there is external friction between the newly formed layer and the previous one, then cohesive bonds form between the layers, and partial plastic deformation occurs. These effects lead to local heat generation between the layers and reheating of the first transfer layer, which enhances grain growth at its boundary. This is the way a gradient structure is formed in the transfer layer.

It was also found that the grain size grows in the whole layer with increasing plunge force (Figure 5a). The grain growth is due to an increase in heat input during FSW in accordance with the classical heat release equation [15]. The application of ultrasound during processing did not significantly affect the grain size in general, but the average grain size over the layer slightly increased due to ultrasonic treatment.

### 3.2. Scanning Electron Microscopy

Alloy 2024 is heat treatable and therefore contains particles of the secondary phases of different compositions and states. Some of the most easily identified particles are coarse intermetallics, which contain metallurgical impurity, such as iron, manganese and silicon, as well as incoherent S- (Al_2_CuMg) and θ-phase (Al_2_Cu) particles. Larger and elongated particles are located at grain junctions; small equiaxed particles are most often located inside the grains. As was shown in previous papers [16] and other works [17], the volume fraction of these particles is often inversely proportional to the volume fraction of coherent and semi-coherent particles due to the limited amount of alloying elements in the alloy. The incoherent particles appear as bright contrast in the SEM image (Figure 6). The image was obtained from the same area as in Figure 4b, showing the entire transfer layer. As in the case with the grain structure, there is no visual difference between the transfer layer boundaries and its center, but a quantitative analysis of the volume fraction and sizes of secondary particles shows a relationship.

Figure 7 shows the dependences of the size and volume fraction of secondary particles on their position in the transfer layer. The measurements were carried out with a large step to improve the measurement statistics, but nevertheless the curves show a particular behavior. As in the case with the grain structure, the volume fraction and average size of particles decrease towards the center of the layer. The growth of secondary particles is a thermally activated process; therefore, such a gradient structure is associated with specific heat release at the layer boundaries. Repeated heat release at the boundary of the first transfer layer enhances the precipitation of secondary particles.

Ultrasonic treatment causes a decrease in the volume fraction and average size of particles as a result of their strain-induced dissolution due to the acoustoplastic effect. This effect was discovered in the studied alloy in previous works [16]. The acoustoplastic effect in the context of friction stir welding implies more intense deformation under ultrasonic vibrations. More intense deformation in turn enhances the strain-induced dissolution of secondary particles.

### 3.3. Transmission Electron Microscopy

Figure 8 shows the TEM images of the foil from the stir zone of the FSW joint. This area was in the center of the transfer layer. Particles of secondary phases were observed in bright-field images. There are predominantly rod-shaped precipitates and rounded particles. The interpretation of diffraction patterns revealed that all secondary and tertiary precipitates are the S-phase (Al_2_MgCu) of the system Al-Cu-Mg. One group of the rounded S-phase precipitates are larger and reach 1 μm in size. These are secondary S-phase precipitates with the [−201]_S_ zone axis partially dissolved during welding. Another group of S-phase precipitates with the morphology of nanosized thin plates or rods are tertiary precipitates with the [3–10]_S_ zone axis. Here, they are called tertiary as they precipitated again after welding. The particles are dissolved during welding due to deformation and high temperature and then precipitate again upon cooling. The detected particles are semi-coherent.

The studies revealed the presence of such particles throughout the entire thickness of the transfer layer, but in different amounts. For example, Figure 9 shows the results of observations close to the transfer layer boundary. This region exhibits the same particles as in the previous area, but in a much smaller amount. The examination results for the entire transfer layer confirmed that the largest amount of particles is observed in the center of the layer.

A similar picture is observed in the UAFSW joint. The same particles and a similar particle distribution over the layer thickness are detected. For example, Figure 10 shows the bright-field TEM images of the stir zone in different areas. As in the previous specimen, a much smaller amount of particles is observed at the transfer layer boundary than in its center.

Figure 11 shows the curves of the volume fraction of semi-coherent secondary particles versus their position in the transfer layer. According to the calculation results, the particle volume fraction is larger in the center of the layer and lower at the boundaries. This particle distribution is preserved after ultrasonic treatment. As expected, the dependence is inversely proportional to the dependence of the volume fraction of incoherent particles. The more intense is the precipitation of incoherent particles, the smaller is the amount of semi-coherent particles. At first glance, the 10–12% volume fraction of secondary particles seems too high for an alloy with the total impurity content of 7.8%. However, the layer contains on average only 5.4% of secondary particles in the FSW joint and 3.6% in the UAFSW joint. As a result of severe plastic deformation during FSW, the particles dissolve and form a supersaturated solid solution. When particles precipitate from the supersaturated solid solution, impurity atoms migrate and form bonds. These factors can contribute to the nonequilibrium distribution of particles in the layer. This is especially noticeable in TEM images where a completely different amount of particles is observed in neighboring grains. Therefore, for the accuracy of the experiment, the particle volume fraction was measured from several images of one area in order to capture several grains.

### 3.4. Microhardness Measurements

The gradient structure of the transfer layer is also confirmed by microhardness measurements with a small indentation spacing of 50 µm. Figure 12 shows the dependence of microhardness on the position in the layer plotted from two transfer layers for better statistics. One can see that the microhardness in the center of the layer is systematically higher than along the boundaries, which is explained by the dispersion strengthening of secondary particles. As shown earlier, the volume fraction of incoherent particles is smaller and that of semi-coherent particles is larger in the center of the layer than at its boundaries. Correspondingly, the microhardness is higher in the center than along the boundaries. This indicates a gradient of mechanical properties in the transfer layer.

Ultrasound treatment led to a decrease in the average microhardness, despite the fact that large softening particles dissolved under ultrasonic vibrations. Presumably this is due to a decrease in welding stresses, because this effect was also observed by other authors [18]. However, a common behavior is evident.

### 3.5. Structural Evolution Pattern

The study of the grain structure and secondary particles revealed a gradient structure in the transfer layer of FSW joints: the grain size, volume fraction and average size of incoherent particles increase towards the layer boundaries, while the volume fraction of semi-coherent particles decreases. The structure evolves due to deformation and high temperature effects. These effects can be conveniently considered using the grain structure as an example (Figure 13). At the beginning of welding, the tool is plunged into the material with the initial structure consisting of elongated grains. The rotating tool transfers the material from the front to the rear side. Heating and severe plastic deformation induce recrystallization. Solid solution grains are refined and become equiaxed. At the second stage, there is already a transfer layer with a fine-grained structure. The tool at this stage takes another material layer and also transfers it from the front to the rear. In the third stage, the second layer comes into contact with the first one. Friction occurs between the layers, leading to heat generation. The layers cohere to each other and stop moving, and heat is dissipated into both layers. The released heat causes grain growth at the interface between the layers. The phase structure is rearranged in the same way. Incoherent particles first dissolve in the transfer layer as a result of deformation, and when the second layer is transferred, they precipitate again at the layer boundary due to additional heat release. The variation of the particle volume fraction across the layer thickness fully explains the appearance of contrast in optical images. Incoherent particles are mostly etched away during etching with Keller’s reagent. Accordingly, the chemical activity of the material is higher and etching is more intense in regions with a larger amount of particles. Therefore, transfer layers in metallographic images are observed as bands and onion rings.

## 4. Conclusions

This paper explored the transfer layer structure of aluminum alloy 2024 after friction stir welding. The main results that were obtained are the following:

1.It was shown for the first time that the transfer layer has a gradient structure. The grain size, the volume fraction and size of incoherent intermetallic particles decrease towards the center of the layer, while the volume fraction of semi-coherent secondary particles increases.2.The gradient structure of the transfer layer is confirmed by microhardness measurements. The microhardness increases towards the center of the transfer layer.3.The onion ring structure formation during friction stir welding explained by a mass transfer mechanism. According to the mechanism, the tool transfers some material from the front to the rear. The material interacts with the previous layer. Friction occurs between the layers, leading to heat generation. As a result, the microstructure at the layer boundaries changes.4.As the plunge force increases, the grain size also increases due to increasing heat generation. The gradient of structure in the transfer layer is preserved in this case.5.Ultrasonic treatment slightly affected the grain structure, but led to a change in the phase structure. In particular, ultrasound reduced the size and volume fraction of secondary particles across the entire layer thickness. The microhardness of the weld material decreased, probably due to a decrease in residual stresses.

## Figures and Tables

**Figure 1 materials-15-06772-f001:**
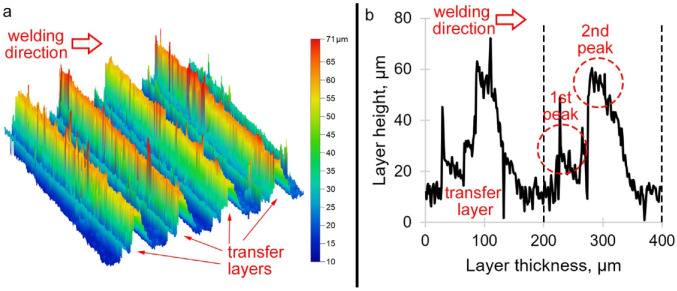
Typical 3D image of the friction surface (**a**) and 2D profile of the transfer layers (**b**).

**Figure 2 materials-15-06772-f002:**
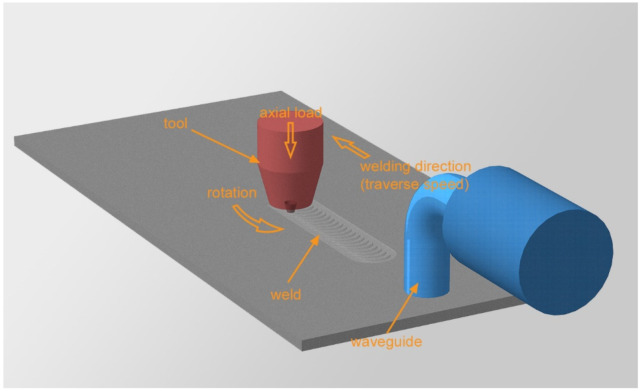
Friction stir welding diagram.

**Figure 3 materials-15-06772-f003:**
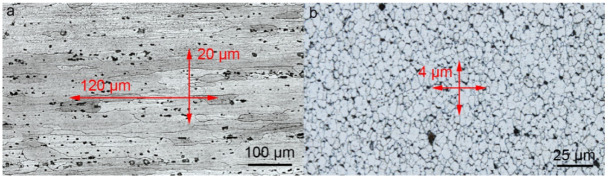
Grain structure of base metal (**a**) and stir zone (**b**).

**Figure 4 materials-15-06772-f004:**
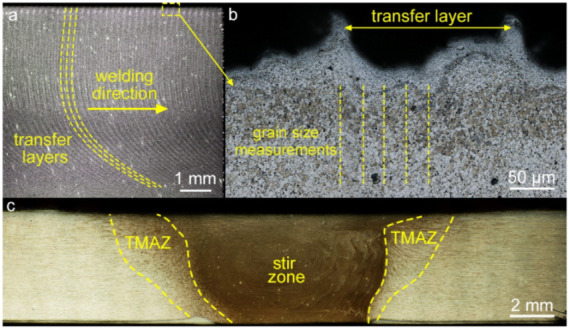
Metallographic images of stir zone in a longitudinal section (**a**) and the same at a higher magnification (**b**) and cross-section image of the specimen (**c**).

**Figure 5 materials-15-06772-f005:**
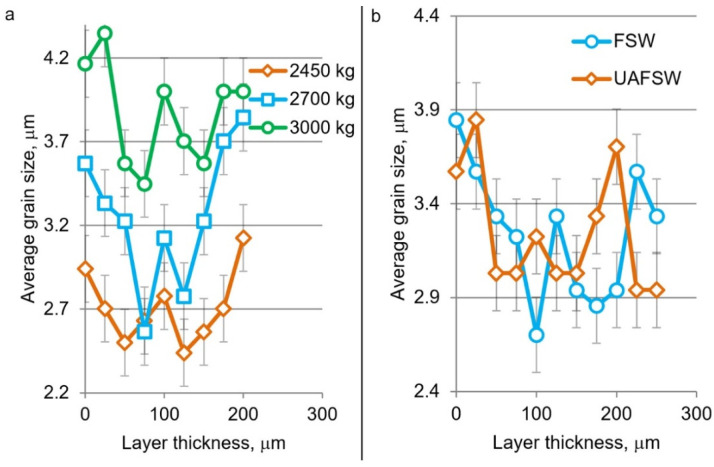
Results of grain size measurements in the transfer layer depending on the axial load (**a**) and at the influence of ultrasound (**b**).

**Figure 6 materials-15-06772-f006:**
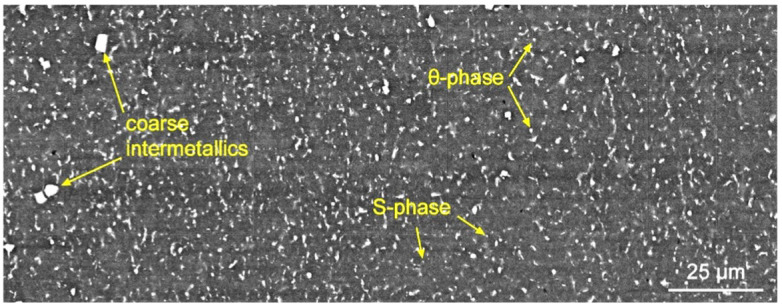
SEM image of stir zone in a longitudinal section.

**Figure 7 materials-15-06772-f007:**
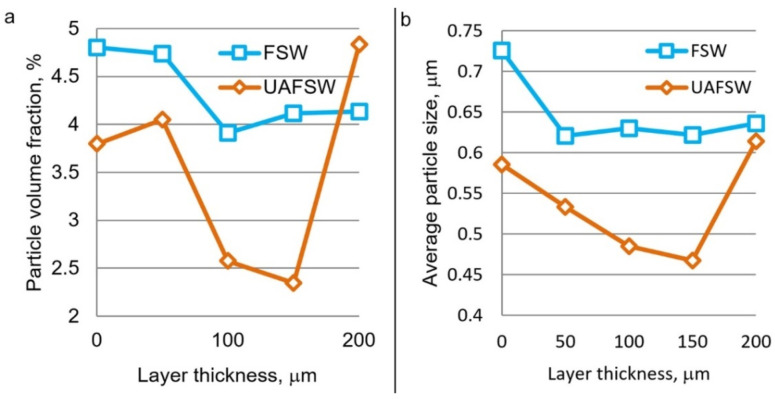
Results of measurement of the volume fraction (**a**) and average size (**b**) of secondary particles in the transfer layer of FSW and UAFSW joints produced at an axial load of 2700 kg.

**Figure 8 materials-15-06772-f008:**
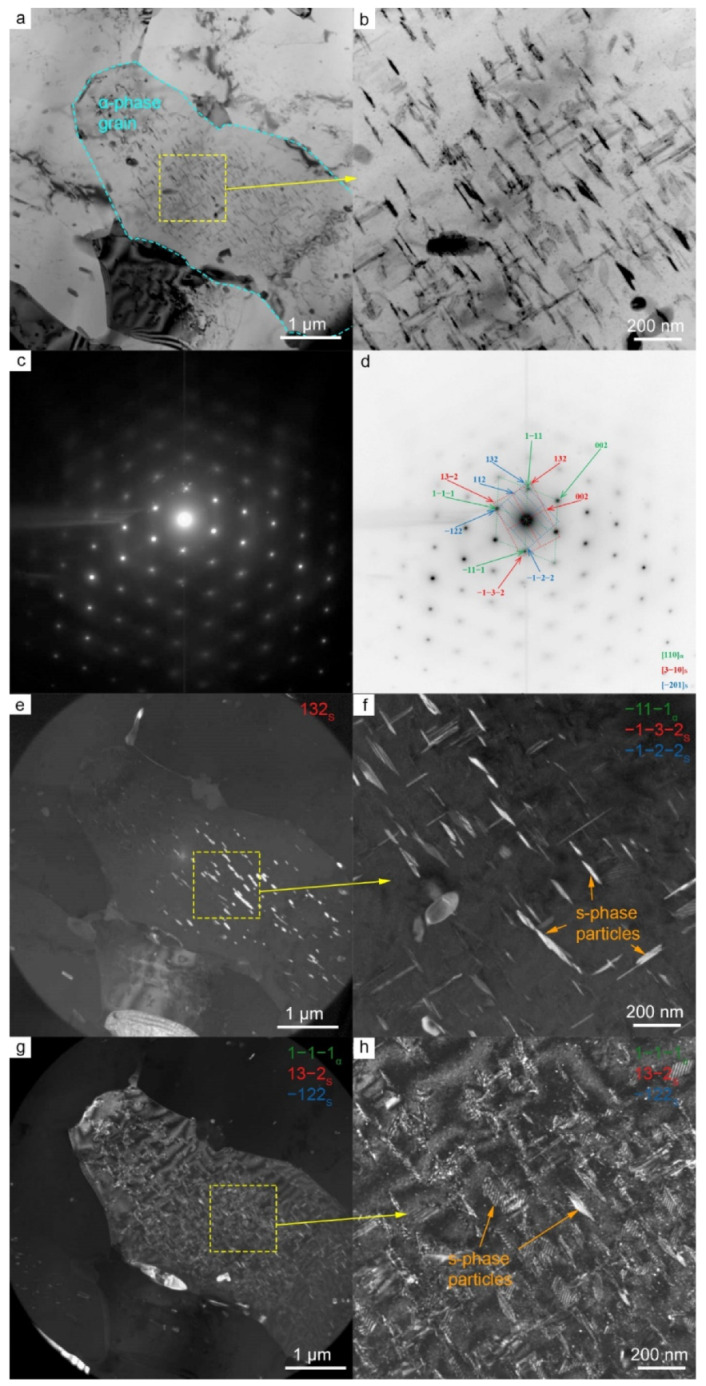
Microstructure of AA2024 in the SZ of FSW joint. Bright-field (**a**,**b**) and dark-field (**e**–**h**) TEM images, SAED (**c**) and SAED pattern indexing (**d**). Dark-field images taken in the reflection 132_S_ (**e**), group of reflections −11−1_α_ & −1−3−2_S_ & −1−2−2_S_ (**f**), and group of reflections 1−1−1_α_ & 13−2_S_ & −122_S_ (**g**,**h**).

**Figure 9 materials-15-06772-f009:**
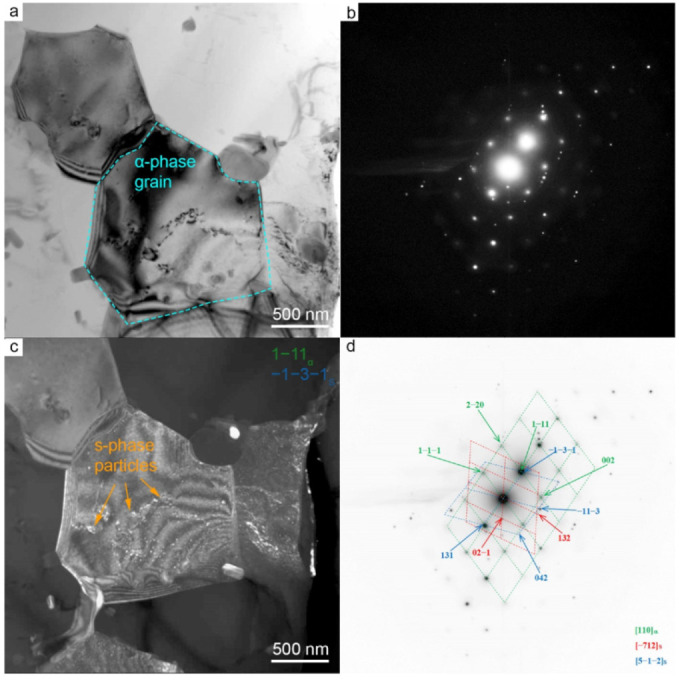
Microstructure of AA2024 in the SZ of FSW joint. Bright-field (**a**) and dark-field (**c**) TEM images, SAED (**b**) and SAED pattern indexing (**d**). Dark-field image taken in the group of reflections: 1−1−1_α_ & −1−3−1_S_.

**Figure 10 materials-15-06772-f010:**
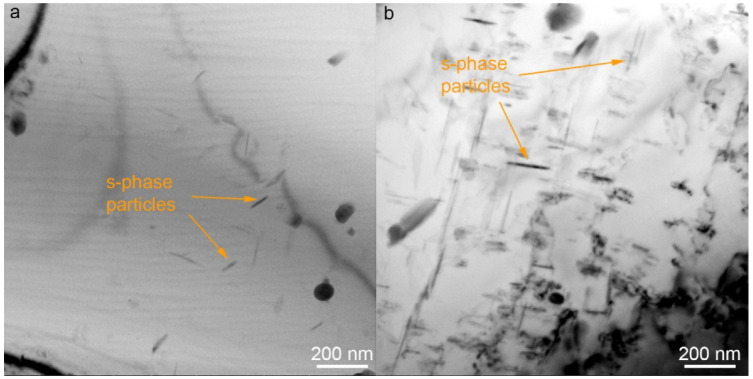
Bright-field TEM images of the UAFSW joint stir zone at the boundary (**a**) and in the center of the transfer layer (**b**).

**Figure 11 materials-15-06772-f011:**
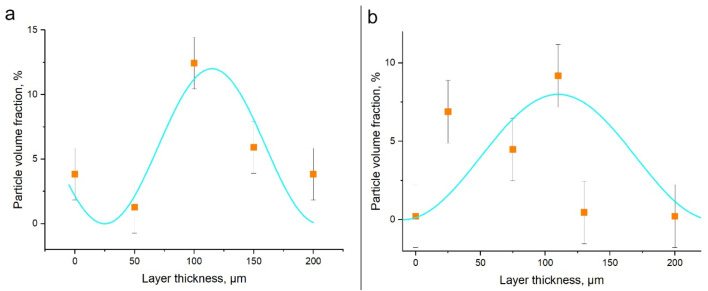
Volume fraction of secondary particles in the transfer layer in the FSW joint (**a**) and UAFSW joint (**b**). The blue color shows an approximation by a sinusoid.

**Figure 12 materials-15-06772-f012:**
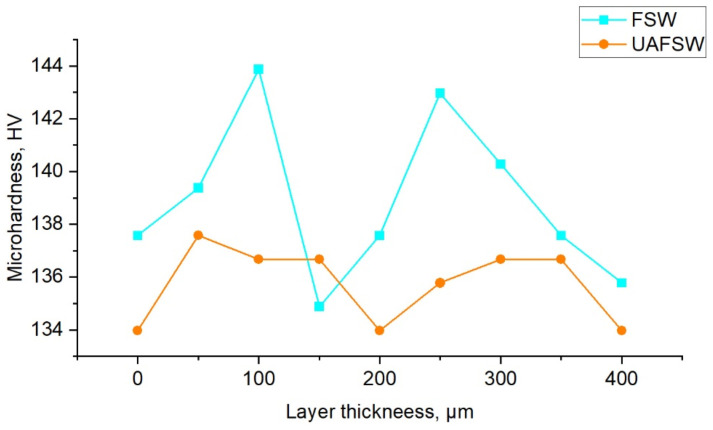
Microhardness line graph of FSW and UAFSW joints.

**Figure 13 materials-15-06772-f013:**
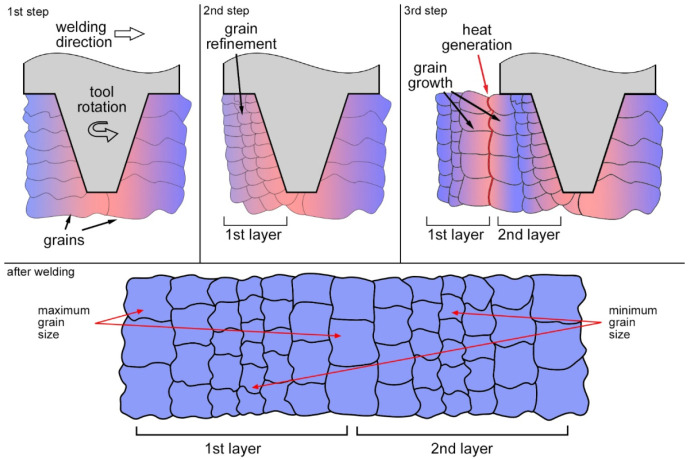
Transfer layer recrystallization scheme and a model of gradient structure in a longitudinal section.

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
