# Peer review of "Gradient Structure of the Transfer Layer in Friction Stir Welding Joints"

_materials, 2022, doi:10.3390/ma15196772_

Round 1

Reviewer 1 Report

The work of this paper is of interest, but the manuscript needs some work for publication.

1. Please give cross-section image of the specimen indicated the weld zone;

2. Please draw a model of gradient structure for FSW process

3. Please give a deeply study on the mechanism of layer transfer during FSW process.

4. The conclusion need complete revision.

Author Response

The work of this paper is of interest, but the manuscript needs some work for publication.

  1. Please give cross-section image of the specimen indicated the weld zone;

Image added in Figure 4 c.

  1. Please draw a model of gradient structure for FSW process

Image added in Figure 13.

  1. Please give a deeply study on the mechanism of layer transfer during FSW process.

It seems to us that, so far, this has been best investigated in the following works:

  • Tarasov, S.Yu.; Filippov, A.V.; Kolubaev, E.A.; Kalashnikova, T.A. Adhesion Transfer in Sliding a Steel Ball against an Aluminum Alloy. Tribol Int 2017, 115, 191–198, doi:10.1016/j.triboint.2017.05.039.
  • Yang, C.L.; Wu, C.S.; Lv, X.Q. Numerical Analysis of Mass Transfer and Material Mixing in Friction Stir Welding of Aluminum/Magnesium Alloys. J Manuf Process 2018, 32, 380–394, doi:10.1016/j.jmapro.2018.03.009.
  • Kalashnikova, T.; Chumaevskii, A.; Kalashnikov, K.; Fortuna, S.; Kolubaev, E.; Tarasov, S. Microstructural Analysis of Friction Stir Butt Welded Al-Mg-Sc-Zr Alloy Heavy Gauge Sheets. Metals (Basel) 2020, 10, 806, doi:10.3390/met10060806.

Added in the Introduction.

  1. The conclusion need complete revision.

Unfortunately, we didn't get the essence of the comment, but we tried to rephrase a few things.

Thank you for your comments.

Reviewer 2 Report

Comments for revision:

(1)   In Section 2.1, the diameter of ultrasonic horn should be given out.

(2)   In page 5, there are “S- and θ-phase particles”, when S- and θ-phase first appear, their chemical formula should be given.

(3)   What is the coarse intermetallics in Figure 6?

(4)   How is the data in Figure 11 obtained? It shall be explained in the corresponding position in the text.

Author Response

Comments for revision:

(1) In Section 2.1, the diameter of ultrasonic horn should be given out.

The diameter of the ultrasonic waveguide was 20 mm. Added in Section 2.1.

(2) In page 5, there are “S- and θ-phase particles”, when S- and θ-phase first appear, their chemical formula should be given.

The formulas are given.

(3) What is the coarse intermetallics in Figure 6?

Сoarse intermetallics is a secondary particles, which contain metallurgical impurity, like iron, manganese, and silicon. Described in Section 3.2.

(4) How is the data in Figure 11 obtained? It shall be explained in the corresponding position in the text.

Microstructural studies were carried out using a JEM-2100 (JEOL Ltd., Akishima, Japan) transmission electron microscope. The foils for the study were prepared by ion thinning using a JEOL EM-09100IS ion slicer. The samples were cut out from the weld in the longitudinal direction. To increase the observation area, the ion beam was inclined at an angle to the sample surface. Thus, the length of the area was about 200 µm and we could obtain images from all points of the transfer layer. The volume fraction of semi-coherent particles was measured by the planimetric method on dark-field images along this area. Thus, it was possible to measure the volume fraction of particles in different regions of the transfer layer and to draw the dependence of the volume fraction on the location in the layer. Added in Section 2.2.

Thank you for your comments.

Reviewer 3 Report

The authors presented a very good and interesting work. This findings may further enhance mostly on the application of FSW. 

The paper was well organized and the reader can follow the articles to the end.

The presentation of the figures are very good. 

The problem was clearly stated.

Better to add why and how you apply the ultrasound. What is the advantages of this device to the FSW?

Figure 2 is too crowded and better to have a schematic diagram to represent the setup.

However, most of the references are outdated. Few add more latest research article relevant to the topic discussed.

The claimed at the first sentence in the abstract and introduction is too much to me. Unless evidence given here in the paper.

Please avoid the use of 'we', 'our' etc. in the manuscript.

Author Response

The authors presented a very good and interesting work. This findings may further enhance mostly on the application of FSW.

The paper was well organized and the reader can follow the articles to the end.

The presentation of the figures are very good.

The problem was clearly stated.

  1. Better to add why and how you apply the ultrasound. What is the advantages of this device to the FSW?

Ultrasound was transmitted into the workpiece across a bolted connection through a magnetostrictive transducer. The ultrasound frequency was 22.1 kHz; the power was 1 kW. The diameter of the ultrasonic waveguide was 20 mm. Ultrasonic exposure due to the acoustoplastic effect can improve the stirring of the material without significantly heating it. In this way, the quality of the joints can be improved. In the context of this work, this is not so significant. However, it was found in previous work that ultrasound can affect the thickness of the transfer layer, so it is used in this work to study in detail the ultrasound effect on the transfer layer structure. Added in Section 2.1.

  1. Figure 2 is too crowded and better to have a schematic diagram to represent the setup.

The figure has been changed.

  1. However, most of the references are outdated. Few add more latest research article relevant to the topic discussed.

Works added.

  1. The claimed at the first sentence in the abstract and introduction is too much to me. Unless evidence given here in the paper.

We suppose that in every scientific field there are both successes and blind spots. In the abstract, we focus on the unknowns so far. Unfortunately, in the literature we did not find an answer to the question of why layer boundaries are visible. That is why the study was conducted. Proof of data absence is hardly possible. At the beginning of the introduction, we listed the successes that have been achieved by other researchers as a little historical background.

  1. Please avoid the use of 'we', 'our' etc. in the manuscript.

Revised.

Thank you for your comments.